# Comparison and Analysis of the Genomes of Three Strains of *Botrytis cinerea* Isolated from Pomegranate

**DOI:** 10.3390/microorganisms13071605

**Published:** 2025-07-08

**Authors:** Alberto Patricio-Hernández, Miguel Angel Anducho-Reyes, Alejandro Téllez-Jurado, Rocío Ramírez-Vargas, Andrés Quezada-Salinas, Yuridia Mercado-Flores

**Affiliations:** 1Laboratorio de Aprovechamiento Integral de Recursos Bióticos, Universidad Politécnica de Pachuca, Carretera Ciudad Sahagún-Pachuca Km. 20, Ex-Hacienda de Santa Bárbara, Zempoala 43837, Hidalgo, Mexico; patriciofcqb@gmail.com (A.P.-H.); anducho@upp.edu.mx (M.A.A.-R.); alito@upp.edu.mx (A.T.-J.); rocioramirez@upp.edu.mx (R.R.-V.); 2Servicio Nacional de Sanidad Inocuidad y Calidad Agroalimentaria, Dirección General de Sanidad Vegetal, Carretera Federal México-Pachuca Km 37.5, Tecámac 55307, Estado de México, Mexico; andres.quezada.i@senasica.gob.mx

**Keywords:** gray mold, phytopathogen, pomegranates, genome analysis, fungi

## Abstract

Gray mold disease, caused by the fungus *Botrytis cinerea*, affects a wide variety of plants. In this study, we conducted several in vitro tests and genomic analyses on three strains of this fungus (BcPgIs-1, BcPgIs-3, MIC) previously isolated from diseased pomegranate fruits, collected at two geographic locations in Mexico. Our goal was to identify possible differences among these strains. The development of the three strains in distinct culture media, the production of extracellular enzymes, and their effect on the progression of infection in pomegranate fruits were evaluated. The genomes were sequenced using the Illumina platform and analyzed with various bioinformatics tools. All strains possess genetic determinants for virulence and cell wall polymer degradation, but MIC exhibited the highest pectinolytic activity in vitro. This strain also produced sclerotia in a shorter time (7 days) in PDA medium. BcPgls-3 demonstrated the highest conidia production across all the culture media used. Both BcPgls-3 and MIC damaged all the pomegranate fruits 8 days after inoculation, while the BcPgls-1 required up to 9 days. Sequencing of the three strains yielded high-quality sequences, resulting in a total of 17 scaffolds and genomes that exceed 41 million bp, with a GC content of approximately 42%. Phylogenomic analysis indicated that the MIC strain is situated in a group separate from BcPgIs-1 and BcPgIs-3. BcPgIs-3 possesses more coding sequences, but MIC has more genes for CAZymes and peptidases. The three strains share 10,174 genes, while BcPgIs-3 and MIC share 851. These findings highlight the differences among the strains studied, which may reflect their adaptive capacities to their environment. Results contribute to our understanding of the biology of gray mold in pomegranates and could assist in developing more effective control strategies.

## 1. Introduction

The pomegranate is the fruit of the pomegranate tree (*Punica granatum* L). Its cultivation originated in Central Asia [1], but it is distributed in numerous areas of the world, including South Africa, North Africa, the Middle East, Israel, Australia, North America, and South America [2]. Production, however, faces numerous phytosanitary challenges, with fungal diseases being the main obstacles due to their significant impact on productivity [3]. Gray mold disease can affect pomegranates, and its presence has been reported in Greece and Mexico [4,5]. The symptoms of this disease include watery lesions with gray mycelium that penetrate and completely invade the interior of the fruit, which may fragment and mummify, though it can remain on the tree, serving as a source of inoculum that can be spread by the wind [5,6].

The disease is caused by *Botrytis cinerea*, a necrotrophic fungus that belongs to the ascomycete group. It is distributed worldwide and affects a significant number of plants with reports of its establishment in 1606 species, causing considerable economic losses worldwide [6,7,8]. The life cycle of gray mold comprises both asexual and sexual stages. In the former, conidiophores are produced, which release mature conidia that serve as the primary inoculum, that is, the main means of spreading the disease. There is also a secondary inoculum, the sclerotia, which are melanized resistance structures that remain viable for many years and are produced during winter or under favorable conditions. During the sexual phase, microconidia can fertilize the sclerotia, forming an apothecium where ascospores are generated [8]. This phytopathogen has advanced mechanisms that enable it to both invade and evade the host’s defense responses. Infection begins with the germination of conidia on the surface of plant tissues. These structures are produced in conidiophores that develop from infected plant tissues or when the sclerotia germinate. Penetration can follow directly through stomata and wounds. Additionally, the germ tube can differentiate into an appressorium or infection cushion, where it secretes phytotoxic metabolites, reactive oxygen species, and enzymes that degrade the plant cell wall. Colonization of host tissues then occurs, followed by sporulation and the production of new conidia [8,9,10].

*B. cinerea* has a complex genomic structure characterized by multiple chromosomes, a broad range of genes associated with pathogenicity, and significant genetic diversity shaped by climate, geography, and plant host variation [8,11]. The generalist, non-specific nature of this pathogen can be attributed to its high virulence and the involvement of numerous genes that regulate infection across different hosts [12]. In this context, genomic research not only aids in understanding the biology of plant pathogens but is also crucial for establishing the groundwork for more effective disease management and control strategies in agriculture.

The aim of this study was to compare three strains of *B. cinerea* isolated from pomegranate orchards in two geographical locations in central Mexico to identify possible differences among them through in vitro testing and genome sequence analysis.

## 2. Materials and Methods

### 2.1. Fungal Strains

The three strains of *B. cinerea* used were BcPgIs-1, BcPgIs-3, and MIC, all isolated from pomegranates with signs and symptoms of gray mold. For the first two strains, fruits were collected from pomegranate orchards located in the State of Mexico [5], while for the third, the collection was carried out in the municipality of Chilcuahutla, Hidalgo, Mexico [13].

To preserve the strains, spore suspensions were prepared in 10% glycerol in 2 mL cryotubes at a concentration of 1.6 × 10^5^ conidia/mL. For reactivation, 10 μL of suspension were placed in the center of a Petri dish with PDA (potato dextrose agar) and incubated at 23 °C for 15 days.

### 2.2. Evaluation of B. cinerea Development in Various Culture Media

Growth of the *B. cinerea* strains under study was evaluated in distinct culture media by preparing pomegranate infusion medium (PIM) plates as follows. An amount of 250 g of fruit (rind, pulp, seeds) was placed in 1 L of boiling distilled water for 30 min. The resulting suspension was allowed to cool to room temperature and then filtered. pH was adjusted to 5.6 with NaOH 1N, 15 g of bacteriological agar was added, and the volume was adjusted to 1 L and sterilized. A minimal medium with pectin (MMP) was prepared with the following composition: 2% yeast extract, 6 g/L NaNO_3_, 1.5 g/L KH_2_PO_4_, 0.5 g/L KCl, 0.5 g/L MgSO_4_·7H_2_O, 0.01 g/L FeSO_4_, 0.01 g/L ZnSO_4_, and 15 g/L agar, with 0.5% pectin added. A variant of this medium, called minimal medium with pectin and pomegranate (MMPP), was also prepared by replacing the water with a pomegranate infusion. Potato dextrose agar (PDA) served as the control.

Inoculation was carried out by placing a 0.5 cm diameter portion of the mycelial growth in the center of a Petri dish that contained the aforementioned media for each fungus analyzed. Incubation conditions were maintained at 23 °C for 10 days, then the morphology of the colonies was documented.

To evaluate sporulation in each culture medium, conidia were harvested from the Petri dishes after 10 days of incubation by adding 15 mL of sterile 0.9% NaCl solution. The conidia were then detached by scraping them with a slide. The suspension was collected using a pipette and transferred to a Falcon tube. Finally, the conidia were counted in a Neubauer chamber. Each assay was performed in quadruplicate [14].

Radial growth kinetics were also conducted in PDA. In this step, a portion of mycelium was inoculated by puncturing one end of the agar plate. The plates were incubated at 23 ± 2 °C for 10 days. Mycelial growth was measured daily and recorded in centimeters.

### 2.3. Production of Enzymes Involved in the Degradation of Plant Cell Wall Polymers

Production of the extracellular enzymes cellulase, pectinase, and laccase by the *B. cinerea* strains was determined in a minimal medium (NaNO_3_ 6 g/L, KH_2_PO_4_ 1.5 g/L, KCl 0.5 g/L, MgSO_4_·7H_2_O 0.5 g/L, FeSO_4_ 0.01 g/L, ZnSO_4_ 0.01 g/L, agar 15 g/L), supplemented with the different substrates of the enzymes to be evaluated at a contraction of 1% (carboxymethylcellulose, pectin, and ABTS [2,2′-azino-bis(3-ethylbenzothiazoline-6-sulfonic acid)] for cellulase, pectinase, and laccase, respectively).

Each fungus was inoculated in the center of the plate. Incubation conditions were 23 °C for 15 days. Enzyme activity was revealed as follows. For cellulase, 5 mL of a 0.2% Congo red solution was added and left to incubate for 30 min. Rinses were then performed with a 1 M NaCl solution until a clear halo appeared around the colony. This was interpreted as a positive result [15]. To determine pectinolytic activity, 5 mL of a 1% hexadecyltrimethylammonium bromide solution was added and stirred gently until a clear halo formed around the colony, again indicating a positive result [16]. In contrast, a green halo around the colony indicated laccase activity production [17]. All experiments were conducted in quadruplicate.

The diameter of the activity halos and fungal colonies was measured using ImageJ v.1.8.0 software. The potency index (PI) for each assay was calculated using the following formula: PI = diameter of the activity halo (cm)/diameter of the fungal colony (cm).

### 2.4. Evaluation of Disease Progression Caused by B. cinerea Strains in Pomegranate Fruits

In this stage of the study, the strains were analyzed to measure their effect on the percentage of disease progression according to the diagrammatic scale reported by Patricio-Hernández et al. [18]. For this purpose, healthy, ripe fruits were selected, washed with soap and tap water, and disinfected by immersion in a 1% sodium hypochlorite solution, followed by two rinses with sterile distilled water and drying with sterile absorbent paper. A 0.5 cm^2^ wound was made at the base of the calyx with a sterile scalpel, and a portion of the mycelium of each fungus was inserted. The inoculated fruits were incubated in a humid chamber at 23 ± 2 °C and monitored daily until they displayed symptoms of rot on their entire surface. All experiments were performed in quadruplicate.

### 2.5. Statistical Analyses

Based on the results from tests on spore production, radial growth kinetics, enzyme production, and disease progression assessment, the relevant statistical analyses were conducted using ANOVA and Tukey’s test with a *p*-value < 0.05. 2.6. Extraction and Sequencing of Genomic DNA

To extract genomic DNA from each strain, mycelium was collected from a 10-day-old culture grown on PDA plates and placed in a sterile mortar. Liquid nitrogen was added to cover the biological material, followed by 2 mL of a lysis solution containing 100 mM Tris-HCl (pH 8), 20 mM EDTA (pH 8), 10 mM NaCl, 1% polyvinylpyrrolidone (PVP), and 2% cetyltrimethylammonium bromide (CTAB). This mixture was ground for 3 min, then 1.0 mL of the mixture was transferred to an Eppendorf tube, which was incubated at 85 °C for 25 min before being placed in an ice bath for 10 min. Using the same tube, an extraction was performed by adding 500 µL of a phenol/chloroform/isoamyl alcohol (25:24:1) solution. This mixture was shaken vigorously and then centrifuged at 14,000 rpm for 15 min. The aqueous phase was collected carefully and transferred to a new tube, where 5 µL of RNase A (10 mg/mL) were added. This new mixture was shaken and incubated at 37 °C for 15 min, then a second extraction was performed using 500 µL of phenol/chloroform/isoamyl alcohol (25:24:1), as described above. The aqueous phase was again transferred to a new tube, and DNA precipitation was initiated by adding a volume of cold absolute ethanol, followed by gentle shaking and centrifugation at 14,000 rpm for 15 min. Two washes with 70% ethanol were then performed. After the ethanol was evaporated at room temperature, the DNA pellet was resuspended in 100 µL of nuclease-free water. DNA integrity was assessed by electrophoresis in a 1% agarose gel. Quality and quantification were determined using a NanoDrop spectrophotometer (Thermo Fisher Scientific Inc, Waltham, MA, USA).

Genome sequencing was conducted for each strain using the Novogene service on an Illumina platform, utilizing 150 bp paired-end reads. DNA libraries were prepared with the Illumina NEBNext Ultra II DNA Library Prep Kit.

### 2.6. Data Quality Control and Genome Assembly

Raw data quality control was performed using FastQC v0.12.0 http://www.bioinformatics.babraham.ac.uk/projects/fastqc/ (accessed on 15 December 2023). Adapters and low-quality sequences were removed with Trimmomatic v-0.39 https://github.com/usadellab/Trimmomatic (accessed on 8 January 2024) [19]. De novo assembly was performed using Spades v3.13.1 considering k-mers of 21, 33, 55, and 77 bp https://github.com/ablab/Spades (accessed on 25 January 2024) [20]. The resulting scaffolds were concatenated in Ragout v2.3 https://github.com/mikolmogorov/Ragout (accessed on 10 May 2024) [21] using the genome of the *B. cinerea* strain B05.10 deposited in the GenBank database, with access code GCF_000143535.2 as a reference. Quast v5.2.0 software https://github.com/ablab/quast (accessed on 15 May 2024)” [22] was used to verify the quality of the assemble. The L50 and N50 values of the assemblies of the three *B. cinerea* strains analyzed were obtained using this software. Finally, a single-copy universal ortholog benchmarking tool (BUSCO v3.0.2) was used with the Fungi Odb10 database to evaluate the integrity of the genome assembly https://anaconda.org/bioconda/busco (accessed on 30 May 2024) [23].

### 2.7. Phylogenetic Analysis

The sequences obtained were compared to the genomes of various *B. cinerea* strains deposited in the GenBank database of the National Center for Biotechnology Information (NCBI) using BLAST software version 2.16.0 https://blast.ncbi.nlm.nih.gov/Blast.cgi (accessed on 2 June 2024). Sequences with similarity values equal to or greater than 98% were selected for alignment on the REALPHY server https://realphy.unibas.ch/realphy/ (accessed on 15 June 2024). Results were used to construct a phylogenomic tree using IQ-TREE version 3.0.1 https://www.hiv.lanl.gov/content/sequence/IQTREE/iqtree.html (accessed on 30 June 2024), which is based on a maximum likelihood bootstrap method and a general time-reversible (GTR) nucleotide substitution model with 1000 replicates and a cutoff of 0.05.

### 2.8. Functional Annotation

Genome annotation was conducted using the Funannotate pipeline https://funannotate.readthedocs.io/en/latest/install.html (accessed on 1 July 2024), while RepeatMasker Version 4.1.9 was employed to identify and eliminate repeated or low-complexity sequences from contigs sized between 250 and 500 bp https://github.com/Dfam-consortium/RepeatMasker (accessed on 1 July 2024) [24]. Gene prediction was conducted using the following tools: Augustus (https://github.com/Gaius-Augustus/Augustus) [25], snap https://github.com/KorfLab/SNAP (accessed on 1 July 2024) [26], glimmerHMM https://github.com/kblin/glimmerHMM (accessed on 1 July 2024) [27], CodingQuarry https://anaconda.org/bioconda/codingquarry (accessed on 1 July 2024) [28], and GeneMark-ES/ET https://genemark.bme.gatech.edu/ (accessed on 1 July 2024) [29]. This process generated multiple inputs and consensus models, which were subsequently filtered using Evidence Modeler https://github.com/EVidenceModeler (accessed on 1 July 2024) [30]. tRNA prediction was conducted with tRNAscan-SE https://github.com/UCSC-LoweLab/tRNAscan-SE (accessed on 1 July 2024) [31]. Results were processed with tbl2asn https://github.com/elucify/genbank-book/blob/master/tbl2asn2.md (accessed on 1 July 2024).

Gene ontology (GO) annotation was conducted using the PANNZER2 web tool http://ekhidna2.biocenter.helsinki.fi/sanspanz/ (accessed on 1 August 2024) [32], while InterProScan https://www.ebi.ac.uk/interpro/download/InterProScan/ (accessed on 10 August 2024) [33] and MEROPS https://www.ebi.ac.uk/merops/ (accessed on 12 August 2024) [34] were utilized to identify functional domains and classify protease families, respectively. The dbCAN3 server https://bcb.unl.edu/dbCAN2/ (accessed on 30 August 2024) was utilized to identify enzymes associated with the degradation, modification, and synthesis of carbohydrates (CAZymes).

The graphical representation of the annotations was done in Circos v.0.69-8 https://circos.ca/documentation/tutorials/quick_start/histograms/configuration (accessed on 13 September 2024).

### 2.9. Comparative Analysis of Coding Sequences

A comparative analysis of the coding sequences from the genomes of the three strains of *B. cinerea* was conducted using the Python version 3.12.2 code specified at https://pypi.org/project/matplotlib-venn/ (accessed on 20 September 2024) which produced a Venn diagram.

### 2.10. Search for Genes Related to Plant Cell Wall Degradation and Pathogenicity

The search for enzymes that can degrade plant cell wall components was conducted using the dbCAN2 server [35]. Enzymes were selected from the HHMER-dbCAN-SUB database in relation to the specific substrates they degrade.

Virulence-related genes were identified based on reports in the literature, then a search was conducted for them in the *B. cinerea* B05.10 genome under the NCBI GenBank access key GCA_000143535.4. Once each gene was identified in the reference genome, it was aligned with the genomes of the three *B. cinerea* strains examined using Clustal Omega software version 1.2.4 [36] to ascertain the position of each one in the genomes of the strains of interest. The identified genes were analyzed using blastp against pathogen–host interaction database (PHI-base) v.5 http://www.phi-base.org/index.jsp (accessed on 23 June 2025) [37].

The search for secondary metabolites phytotoxic was conducted using antiSMASH software fungal version 8.0 https://fungismash.secondarymetabolites.org/#!/start (accessed on 25 June 2025) [38].

## 3. Results

### 3.1. Evaluation of the Development of B. cinerea in Distinct Culture Media

The growth of the three strains analyzed was assessed in various culture media. The MIC strain produced a brown agar-diffusible pigment in three media: MPI, MMP, and MMPP. This strain also showed sclerotia formation seven days after growth in the PDA medium (Figure 1).

For the BcPgIs-1 and BcPgIs S-3 strains, the colonies in all media had a velvety white to beige growth. In contrast, the MIC strain colony presented a white background with some gray areas in PDA. Growth in the MPI and MMP media was equally velvety but dark gray in color. In the MMPP, half of the color was brown with some gray areas (Figure 1).

Regarding conidia production, BcPgIs-3 exhibited the highest capacity for producing these structures, followed by BcPgIs-1. In contrast, MIC showing the lowest conidia production in the MMPP medium (Figure 2).

The evaluation of the development of each strain in PDA revealed that BcPgIs-1 invaded the plate in 8 days; thus, requiring less time than the other two strains, which took until day 9 (Figure 3).

### 3.2. Enzyme Production Associated with the Degradation of Plant Cell Wall Polymers

All strains under study exhibited cellulolytic and pectinolytic activity, including laccase. MIC demonstrated the highest power index for pectinase production. The three strains presented similar values for cellulase and laccase production (Figure 4).

### 3.3. Tests of the Advance of Infection on Pomegranate Fruits

As shown in Figure 5, the strains BcPgIs-3 and MIC exhibited 100% disease progression in the pomegranate fruits within 8 days, whereas BcPgIs-1 achieved this by day 9.

### 3.4. Genome Sequencing and Assembly

The sequencing process for each strain analyzed produced sequences of acceptable quality. A total of 17 scaffolds were obtained for all three strains, with genome sizes exceeding 41 million base pairs (bp) and a GC content of approximately 42%. The numbers of mRNA and tRNA were consistent across all strains. BUSCO analysis indicated that the integrity of the assemblies was satisfactory, with values exceeding 90%. N50 values were 2,607,131 bp, 2,575,794 bp, and 2,589,848 bp for the BcPgIs-1, BcPgIs-3, and MIC strains, respectively. The L50 value was 7 for all strains (Table 1).

The genome sequences were deposited in the GenBank database with the following accession codes: GCA_040571525.1 and GCA_043381465.1 for the BcPgIs-1 and BcPgIs-3 strains, respectively. The MIC strain is currently available as Biosample SAMN43203181.

### 3.5. Phylogenomic Analysis

Phylogenomic analysis of the genomes of the three strains was performed in conjunction with the genomes of five strains of *B. cinerea*, two of *Botrytis fabae*, and one of *Sclerotinia sclerotium* as an external group in the tree. All strains of interest were grouped with *B. cinerea*. BcPgIs-1 and BcPgIs-3 were found to share the same group as they reached a Bootstrap value of 97%, unlike MIC, which was closer to the strains B10.5 and S13 (Figure 6).

### 3.6. Functional Annotation

Table 2 presents the functional annotation results of the genomes for the three strains studied. BcPgIs-3 has the highest number of coding sequences (10,787) and mRNA (10,578), while MIC exhibits the lowest numbers, with 10,668 coding sequences and 10,459 mRNA. All strains contain 209 tRNA genes. The tools InterProScan and pFam identified more genes in BcPgIs-3. Regarding the genes associated with CAZymes and peptidases (MEROPS), MIC had the highest number of coding sequences, with 445 and 334, respectively.

Figure 7 graphically shows the gene ontology (GO) analysis for the three strains. MIC had the highest gene allocation (9168 genes), followed by BcPgIs-3 (9077), and BcPgIs-1 (8065). Turning to the cellular components process, the BcPgIs-3 and MIC strains have a higher representation of genes for the term ribosome. For these same strains, in terms of the molecular function process, a higher representation of genes was observed for methyltransferase activity, a structural constituent of the ribosome, and nucleic acid binding, which was not observed in BcPgIs-1. Through this analysis we determined that the MIC strain had a genes representation associated with the terms endoplasmic reticulum membran and molecular oxygen cellular component, which was not the case with the other two strains.

Circos plots visually represent gene density, GC content deviation, and GC content in each genome, demonstrating collinearity and indicating potential gene duplication events, which were more prevalent in the BcPgIs-1 strain (Figure 8).

### 3.7. Comparative Analysis of Coding Sequences

A comparative analysis of the coding regions in the genomes of the strains revealed that they share a total of 10,174 genes. In this case BcPgIs-3 and MIC strains had the highest number of shared genes (851). Regarding non-shared genes, BcPgIs-3 had 52, MIC had 33, and BcPgIs-1 had 2 (Figure 9).

### 3.8. Search for Genes Related to Plant Cell Wall Degradation and Pathogenicity

Figure 10 illustrates the number of genes that encoded enzymes responsible for degrading plant cell wall components—such as pectin, lignin, cellulose, and xylan—were identified in the genomes of the strains. Note that all of these strains possess the genetic determinants necessary for the degradation of these polymers. The enzymes encoded by the genes identified in the analyzed genomes are listed in the Appendix A. Regarding pectin degradation, the MIC strain has more genes related to this activity (Figure 10). Unlike BcPgIs-1 and MIC, the BcPgIs-3 strain lacks genes encoding alpha-galacturonidase. However, both the BcPgIs-3 and MIC strains have two additional pectinesterase genes (Appendix A). For lignin degradation, strain BcPgIs-3 has the fewest genes (Figure 10), with five fewer laccases than strains BcPgIs-1 and MIC. It also contains an oxidoreductase and two diphenol oxidases, which are absent in the other two strains. Strain BcPgIs-1, unlike the others, is the only one with three galactose oxidases and two oxidases, while strain MIC has one glucose–methanol–choline oxidoreductase and five oxidases (Appendix A). For cellulose degradation, strain BcPgIs-1 has the most genes (Figure 10) and is the only strain containing cellulases and exoglucanases. Strain MIC has the highest number of genes encoding endoglucanases (Appendix A). Additionally, this same strain has the most genes involved in xylan degradation (Figure 10). Strain BcPgIs-3 lacks genes encoding glycoside hydrolase or xylan alpha-1,2-glucuronosidase. Strain BcPgIs-1 contains three genes for 1,4-beta-xylosidase, while strains BcPgIs-3 and MIC have four (Appendix A).

Results for the genes associated with pathogenicity are presented in Table 3, which shows that eight were identified in the reference strain, with orthologs found in the strains under study. Additionally, the blastp analysis against PHI-base confirmed that these genes are linked to pathogenicity and virulence processes (Appendix A).

In the antiSMASH analysis using the current database, no clusters associated with the synthesis of phytotoxic compounds were identified.

## 4. Discussion

*B. cinerea* is considered as one of the most important phytopathogens in agricultural production [6] due to its ability to infect a wide range of crops of commercial interest [46] and its resistance to various antifungal substances [47]. The present study was carried out to describe and compare three strains of *B. cinerea* –BcPgIs-1, BcPgIs-3, and MIC isolated from pomegranate fruits by means of in vitro tests, and then analyze their respective genomes due to their capacity to induce the signs and symptoms characteristic of gray mold [5,13].

The growth of the strains under study in distinct culture media allowed us to determine that the behavior of the MIC strain differs from that of BcPgIs-1 and BcPgIs-3 by, for example, forming sclerotia in less time. Differences in the production of conidia were also observed, depending on the medium in which each fungus was developed. A distinct study of *B. cinerea* used various culture media with glucose, cellulose, and a composition of deproteinized tomato cell wall as carbon sources. Those authors found that these variations triggered alterations in growth rates, hyphal morphology, conidia production, and metabolites [48].

The genome assemblies obtained consisted of 17 scaffolds per strain. L50 and L90 values were identical for all three strains. It is important to note that these parameters evaluate the number of scaffolds required to cover 50 and 90%, respectively, of the total assembly length [49].

Phylogenetic analysis revealed that BcPgIs-1 and BcPgIs-3 are very similar, likely because both were isolated from nearby geographical locations [5]. In contrast, strain MIC, which was isolated from a geographically distinct area in Chilcuahutla, Hidalgo, Mexico, showed a genetic relationship to *B. cinerea* strains B05.10 and S13. The B05.10 strain was isolated from grapes in Germany, while S13 was sourced from tomato fruits in France. The B05.10 strain has been used extensively for genetic and phenotypic studies of this phytopathogenic fungus [50,51]. The study by Plesken et al. [52], that analyzed the genetic diversity of *B. cinerea* populations by multilocus sequencing, found that strains isolated in different hosts from distinct geographic sites shared phylogenetic similarities and are positioned in the same group with the strain B05.10. *B. cinerea*; however, it has innate genetic plasticity that permits the selection of populations that experience multiple levels of adaptation driven by anthropogenic factors, such as the use of fungicides and the system and type of cultivation, aspects that, consequently, shape the population structure [53].

Regarding the annotation of the genomes obtained for these three *B. cinerea* strains, we observed that they share similar genomic characteristics, while BcPgIs-3 and MIC stood out by having more genes annotated in InterProScan and Pfam. These differences from BcPgIs-1 could reveal specific characteristics related to pathogenicity, the environment, or the fungus’ response to treatments applied to control it [54]. Our work showed that these two strains have the ability to invade the fruit more quickly, and that even the MIC strain presents a higher radial growth rate in PDA and greater pectinolytic activity, possibly related to the presence of a greater number of genes that code for this activity, which is essential during the penetration process [55]. Furthermore, the presence of more carbohydrate-active enzyme-related genes (CAZymes) in the MIC strain could be related to a greater capacity to degrade plant cell wall components, suggesting a higher pathogenic potential [56]. Once again, these variations may be due to selective pressures derived from geographic differences and host adaptations [52]. It can be concluded that the MIC strain isolated in Hidalgo demonstrates more virulence-related characteristics than the BcPgIs-1 and BcPgIs-3 strains isolated in the State of Mexico. Additionally, it belongs to a different clade on the phylogenomic tree.

The gene ontology analysis showed the representation of genes associated with biological processes, cellular components, and molecular functions. This process revealed some differences in the three strains that may be reflected in possible variations in their adaptive and metabolic capacities [53]. For example, a greater representation of genes associated with methyltransferase activity was observed in the BcPgIs-3 and MIC strains but not in BcPgIs-2. This finding is particularly significant since DNA methyltransferases play a crucial role in survival during plant–phytopathogenic fungus interaction. In *B. cinerea*, the DNA methyltransferases BcDIM2 and BcRID2 exhibit a synergistic effect on pathogenicity. The double mutant of these proteins showed slow mycelial growth, low spore germination, reduced oxidative stress tolerance, and complete loss of pathogenicity in several hosts, associated with reduced expression of virulence-related genes [57]. The representation of genes associated with the reduction in molecular oxygen in MIC could be related to its ability to produce enzymes—such as peroxidases—to counteract the H_2_O_2_ produced by plants as a defense method against infection, an activity that promotes colonization and progression of the disease caused by this pathogen [58].

These observations highlight that the differences in gene representation obtained from the GO analysis could be related to specific metabolic and adaptation strategies, which are essential for improving our understanding of each strain’s phenotypic variability.

In the Circos diagrams for the three strains studied, peaks distributed along the scaffolds are visible in relation to GC percentage and GC bias. This condition is related to several fundamental biological processes, including the efficacy and efficiency of transcription and translation, the secondary structure of RNA molecules, and epigenetic modifications of DNA [59]. Regarding genetic density, areas with distinct shades can be observed. The darkest ones signal a higher concentration of genes. In this regard, Raffaele and Kamoun [60] mention that several plant pathogenic fungi have genomes marked by a discontinuous distribution of gene density due to the presence of regions rich in repetitions but scarce in genes.

Comparative analysis of coding sequences revealed that the three strains share a significant number of genes; however, some differences are observed, such as the fact that strains BcPgIs-3 and MIC share a greater number of genes, despite being isolated from two different geographical locations. Regarding the number of unique genes, BcPgIs-1 presented only 2, while BcPgIs-3 presented 52. These two strains were isolated from geographically close sites and are phylogenetically related. On the other hand, in the Circos diagrams, greater collinearity was observed in BcPgIs-1. As mentioned above, these variations may result from selective pressures in response to environmental changes or physiological needs, demonstrating their genetic plasticity [53].

The strains under study were found to possess groups of genes related to the degradation of polysaccharides commonly present in the plant wall of plants, such as pectin, lignin, cellulose, and xylan [61].

The xylanases produced by *B. cinerea* may play a dual role during plant infection. On the one hand, they might be recognized by the host, which would activate defense responses; on the other, their secretion could induce apoptosis, thereby activating the necrotrophic phase of the fungus [62]. There are also reports that *B. cinerea* can degrade lignin through production of extracellular laccases that contribute to the development of rot in the tissues of the plants it infects [63].

The production of phytotoxic compounds is a key virulence factor for *B. cinerea*, as it helps the fungus penetrate host tissues and causes cell death. In this study, the current antiSMASH tool was unable to identify any cluster related to the production of these compounds.

This study found that the same orthologous genes linked to pathogenicity were in all three *B. cinerea* strains. However, the differences observed in in vitro tests may result from variations in gene expression at the transcriptional level or regulation by epigenetic modifications, which are crucial in the pathogenicity process [64,65].

In conclusion, the study of these three *B. cinerea* strains isolated from pomegranate fruits highlights the MIC strain due to its high pectinolytic activity and rapid sclerotia formation, while BcPgIs-3 demonstrated greater conidia production. Although all three strains share a considerable number of genes, they exhibit variations in their genomic profiles, that suggest specific adaptations influenced by geographic and environmental factors unique to their isolation sites. These factors can significantly impact the pathogenic potential and environmental persistence of various strains. Such adaptations provide advantages that enable quicker infections, longer survival under adverse conditions, and varied responses to fungicidal treatments.

## Figures and Tables

**Figure 1 microorganisms-13-01605-f001:**
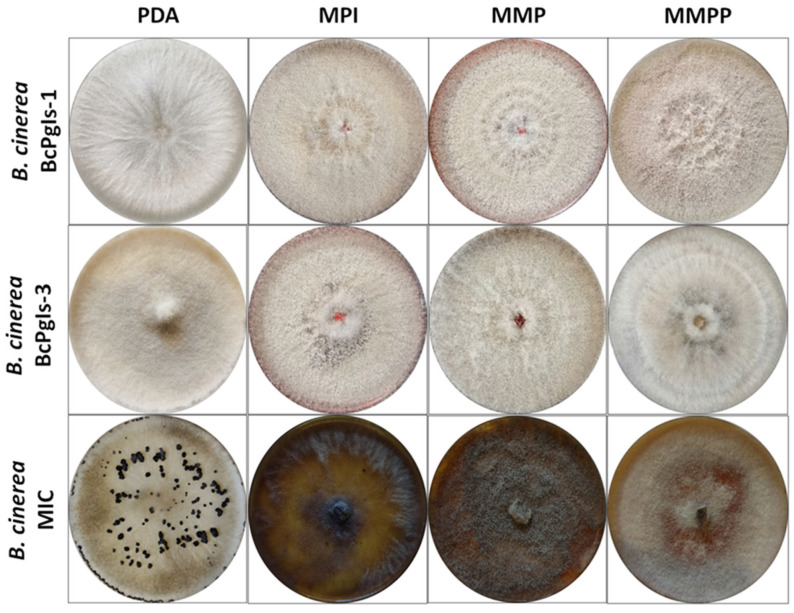
Development of three strains of *B. cinerea* isolated from pomegranate fruits in distinct culture media: PDA (potato dextrose agar), MPI (pomegranate infusion medium), MMP (minimum medium with pectin), and MMPP (minimum medium with pectin and pomegranate infusion). The images shown correspond to the development after 10 days of incubation at 23 ± 2 °C. All tests were conducted with four replicates.

**Figure 2 microorganisms-13-01605-f002:**
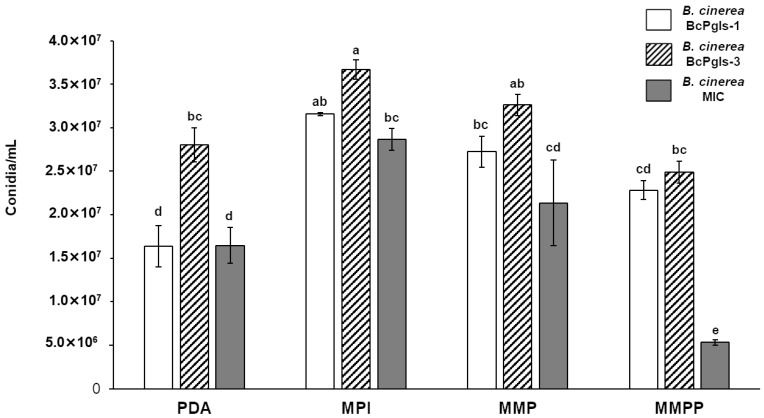
Production of conidia from three strains of *B. cinerea* isolated from pomegranate fruits in distinct culture media: PDA (potato dextrose agar), MPI (pomegranate infusion medium), MMP (minimum medium with pectin), and MMPP (minimum medium with pectin and pomegranate infusion). Data were obtained after 7 days of incubation at 23 ± 2 °C. Letters indicate the results of statistical analyses performed via ANOVA and Tukey’s test (*p* < 0.05). Identical letters indicate differences that were not statistically significant. All tests were conducted in quadruplicate.

**Figure 3 microorganisms-13-01605-f003:**
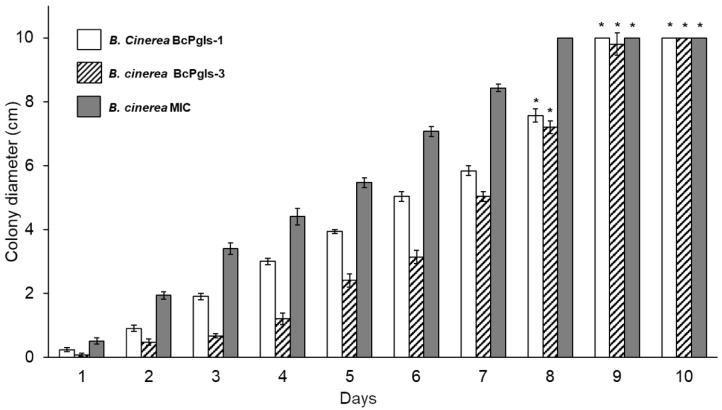
Radial growth of three *B. cinerea* strains isolated from pomegranate fruits on potato dextrose agar (PDA), incubated at 23 ± 2 °C. Asterisks indicate differences that were not statistically significant. Statistical analyses were conducted using ANOVA and Tukey’s test (*p* < 0.05), comparing the three strains for each day.

**Figure 4 microorganisms-13-01605-f004:**
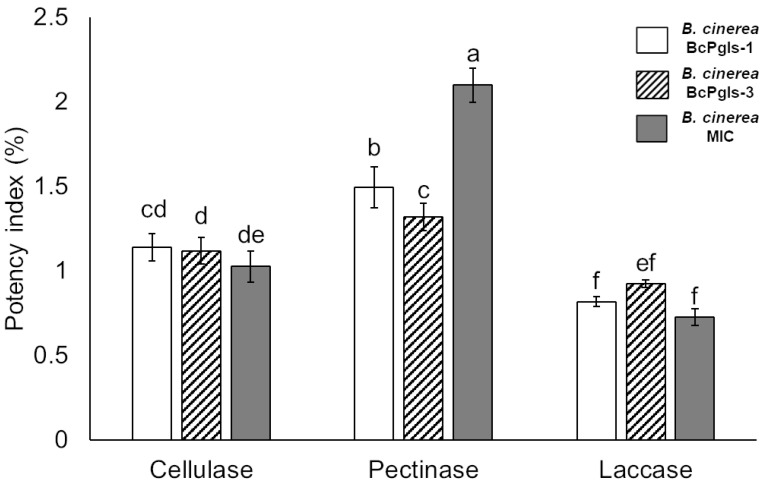
Enzyme production associated with the degradation of vegetable cell wall polymers by three strains of *B. cinerea* isolated from pomegranate fruits. Data were obtained after 15 days of incubation at 23 ± 2 °C. Letters indicate the results of statistical analyses performed via ANOVA and Tukey’s test (*p* < 0.05). Identical letters indicate differences that were not statistically significant. All tests were conducted in quadruplicate.

**Figure 5 microorganisms-13-01605-f005:**
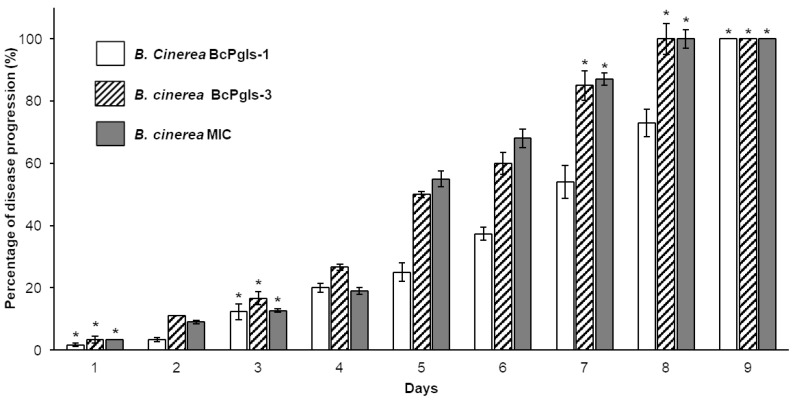
Percentage of disease progression in pomegranate fruits inoculated with distinct strains of *B. cinerea*. The inoculated fruits were incubated in a humid chamber at 23 ± 2 °C. Asterisks indicate differences that were not statistically significant. Statistical analyses were conducted using ANOVA and Tukey’s test (*p* < 0.05), comparing the three strains for each day.

**Figure 6 microorganisms-13-01605-f006:**
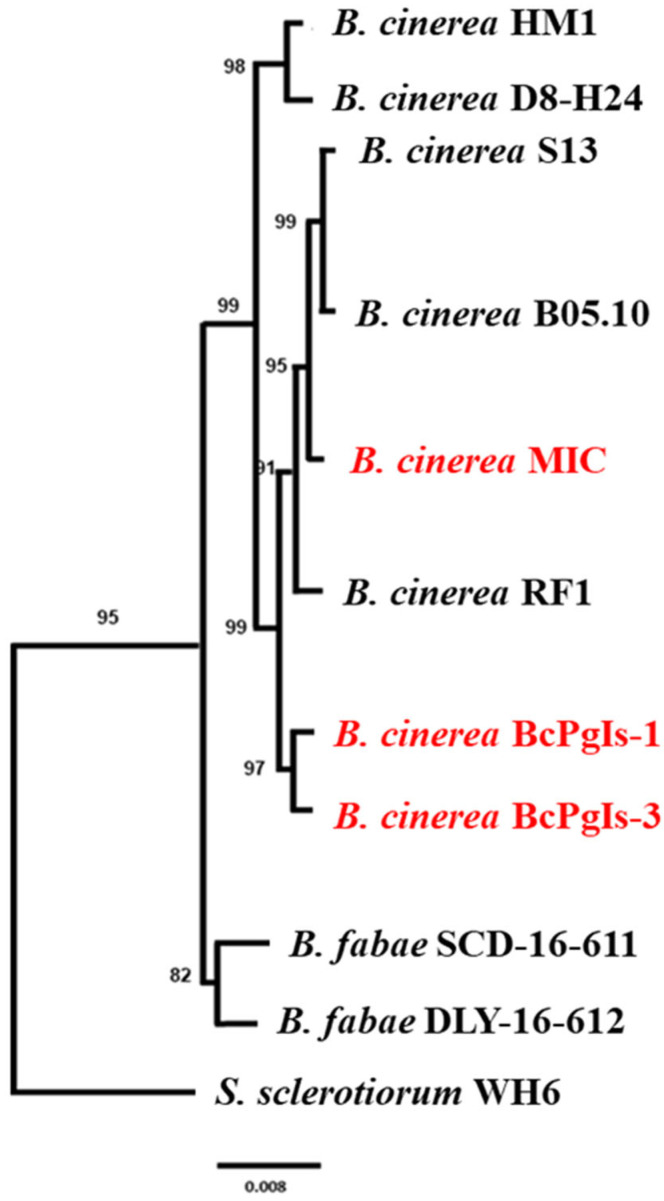
Phylogenomic relationship between *B. cinerea* strains BcPgIs-1, BcPgIs-3, MIC isolated from pomegranate fruits, and other strains of the genus *Botrytis*. The names of the strains being studied are shown in red. The accession codes of the sequences used are as follows: *B. cinerea* HM1; GCA_032197585.1, *B. cinerea* D8_H24; GCA_019186565.1, *B. cinerea* S13; GCA_022560135.1, *B. cinerea* B05.10; GCA_000143535.4, *B. cinerea* MIC; Biosample SAMN43203181, *B. cinerea* RF1; GCA_015147965.1, *B. cinerea* BcPgIs-1; GCA_040571525.1, *B. cinerea* BcPgIs-3; GCA_043381465.1, *B. fabae* SCD-16-611; GCA_004335035.1, *B. fabae* DLY-16-612; GCA_004335055.1, *S. sclerotiorum* WH6; GCA_019022465.1.

**Figure 7 microorganisms-13-01605-f007:**
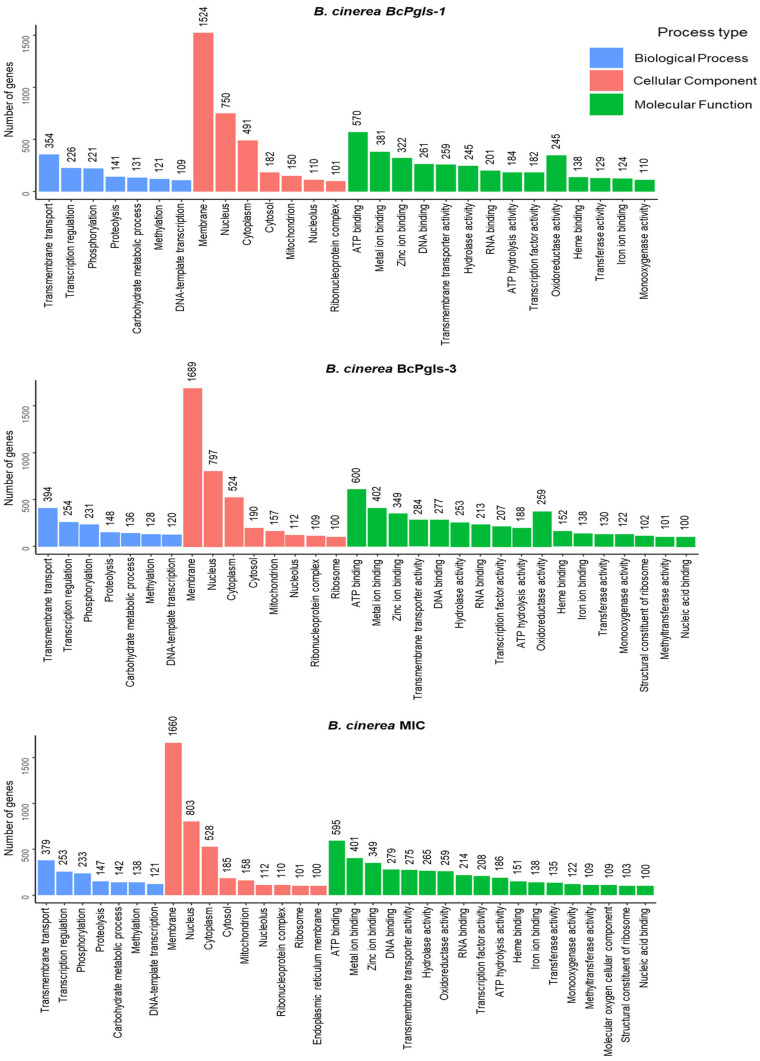
Gene ontology (GO) analysis of the genomes of three *B. cinerea* strains isolated from pomegranate fruit. The number of genes linked to each GO category is shown above each bar.

**Figure 8 microorganisms-13-01605-f008:**
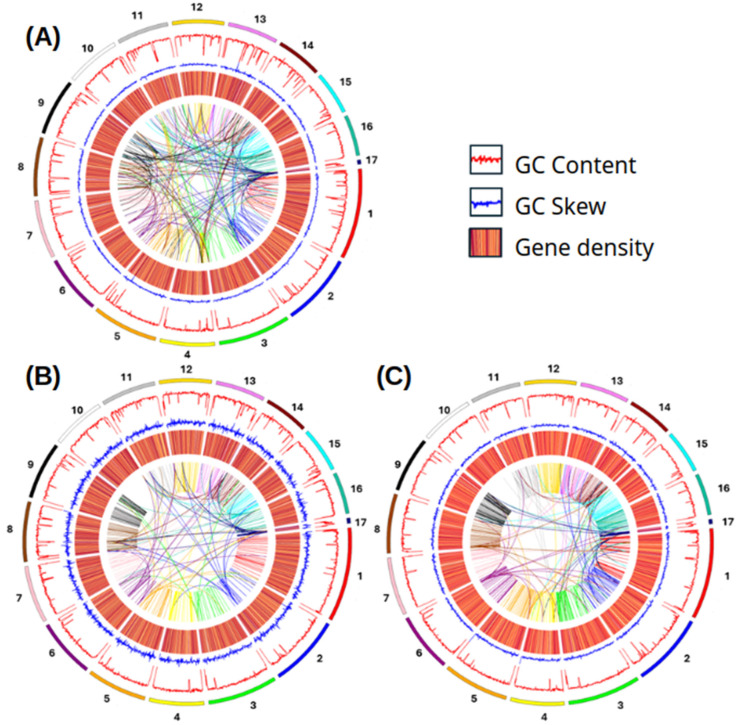
Circos plot of the genomes of *B. cinerea* strains isolated from pomegranate fruits. From inner to outer: the first circle (in the center) is the collinearity analysis; the second is the heat map representing gene density; the third and fourth show GC skewn and GC content, respectively; the fifth is a map of scaffold composition. (**A**) BcPgIs-1; (**B**) BcPgIs-3; (**C**) MIC.

**Figure 9 microorganisms-13-01605-f009:**
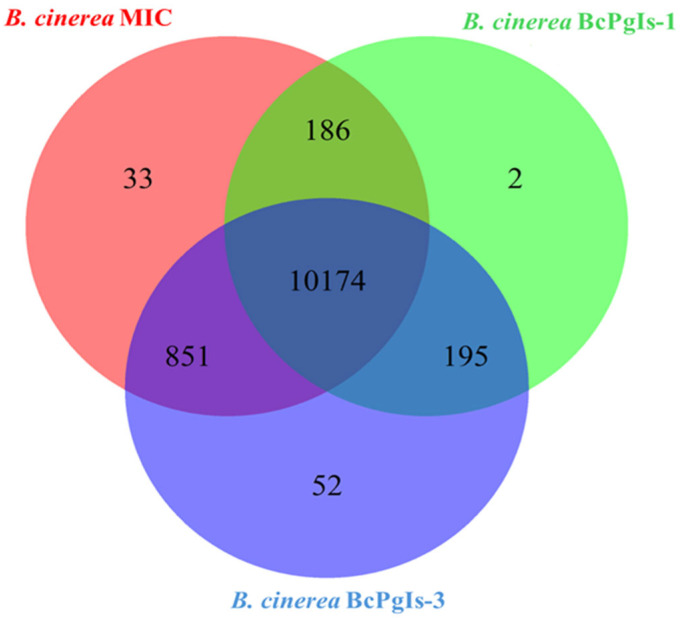
Comparative analysis of the coding regions in the genomes of three *B. cinerea* strains isolated from pomegranate fruits. The Venn diagram illustrates the relationships among the coding regions of the three genomes compared.

**Figure 10 microorganisms-13-01605-f010:**
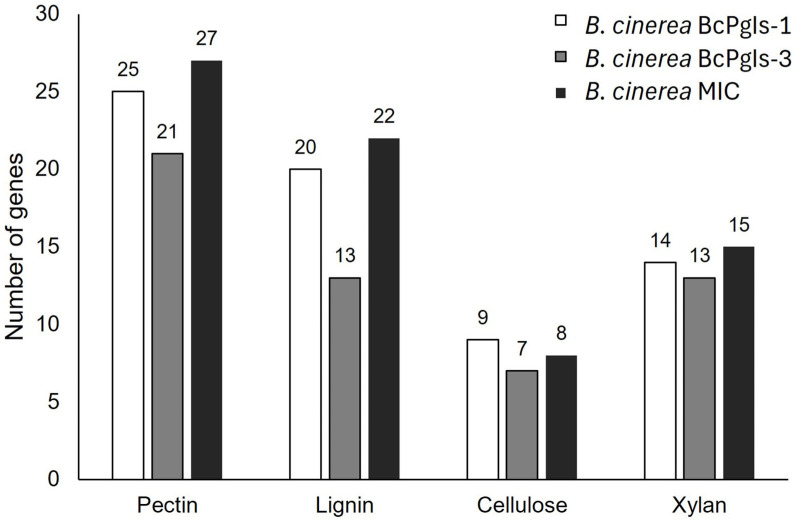
Number of genes encoding enzymes that degrade distinct components of the plant cell wall identified in the genomes of three strains of *B. cinerea* isolated from pomegranate fruits.

**Table 1 microorganisms-13-01605-t001:** Characteristics of the genomes of three *B. cinerea* strains isolated from pomegranate fruits.

Feature	*B. cinerea* Strain		
	BcPgIs-1	BcPgIs-3	MIC
Genome size (pb)	41,776,945	41,672,452	41,417,038
Number of scaffolds	17	17	17
GC content (%)	42.45	42.22	42.48
Shortest scaffolding (pb)	189,773	230,524	208,218
Longest scaffold (pb)	4,215,149	4,076,546	4,062,.245
N50 (pb)	2,607,131	2,575,794	2,589,848
N90 (pb)	2,054,873	2,025,540	2,015,948
L50	7	7	7
L90	14	14	14
BUSCO (%)	93.2	97.0	96.9

**Table 2 microorganisms-13-01605-t002:** Functional annotation of the genomes of three *B. cinerea* strains isolated from pomegranate fruits.

Feature	BcPgIs-1	BcPgIs-3	MIC
Number of coding sequences	10,708	10,787	10,668
mRNA number	10,449	10,578	10,459
tRNA number	209	209	209
Interproscan (number of genes)	7943	8323	8312
PFam (number of genes)	6726	7129	7098
CAZymes (number of genes)	419	435	445
MEROPS (number of genes)	326	330	334

**Table 3 microorganisms-13-01605-t003:** Orthologous genes associated with pathogenicity identified in the genomes of three *B. cinerea* strains isolated from pomegranate fruits.

Stain of *B. cinerea*				Gen	Known Function	Reference
B05.10 (Reference Strain)	BcPgIs-1	BcPgIs-3	MIC			
BCIN_02g02570	FUN_003004-T1	FUN_007008-T1	FUN_008163-T1	*BcATG8*	Autophagy, mycelial development, conidiation, sclerotia formation, and virulence	[39]
BCIN_03g02930	FUN_000057-T1	FUN_002146-T1	FUN_000245-T1	*BcCLA4*	Growth, morphogenesis, conidia production, and pathogenicity	[40]
BCIN_09g06130	FUN_011469-T1	FUN_010843-T1	FUN_010711-T1	*Bcpls1*	Necessary for proper performance of the appressorium	[41]
BCIN_10g05560	FUN_000702-T1	FUN_004250-T1	FUN_002577-T1	*Bcste12*	Transcription factor that controls penetration efficiency	[8,42]
BCIN_02g08170	FUN_001816-T1	FUN_001878-T1	FUN_001831-T1	*Bmp1*	Protein kinases required for surface recognition and host penetration	[43,44]
BCIN_09g02390	FUN_006454-T1	FUN_006459-T1	FUN_006454-T1	*Bmp3*	Protein kinases required for surface recognition and host penetration	[43,44]
BCIN_14g03010	FUN_009511-T1	FUN_009579-T1	FUN_009472-T1	*Bcgbl1*	Regulates protein kinase signaling pathways (Bmp1 and Bmp3)	[44]
BCIN_04g03490	FUN_003110-T1	FUN_003142-T1	FUN_003113-T1	Hypotetical protein	Protein with GAL4-like DNA-binding domain and acetyltransferase activity, which controls key processes such as pathogenicity	[45]

## Data Availability

The original contributions presented in this study are included in the article/Appendix A. Further inquiries can be directed to the corresponding author.

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
