# Peer review of "Comparison and Analysis of the Genomes of Three Strains of Botrytis cinerea Isolated from Pomegranate"

_microorganisms, 2025, doi:10.3390/microorganisms13071605_

Round 1

Reviewer 1 Report

Comments and Suggestions for Authors

Botrytis cinerea is an important necrotrophic fungal pathogen that infects a wide variety of plants and causes gray mold disease. The current manuscript reported a comparison and analysis of the genomes of three strains of B. cinerea isolated from pomegranate. The results obtained could be useful information for researchers who worked in the similar field. In general, this is an interesting study.

Some suggestions are as follows:

  1. Section 2.3. Generally, when measuring enzyme activity, it is necessary to define enzyme units. For example, one enzyme unit may be defined as the amount of enzyme that produces a certain quantity of product per minute under specific conditions, and so on. The “activity halo” can provide a qualitative assessment of enzyme activity, but it is not accurate for quantitative analysis. The authors should recalculate the enzyme activity.
  2. Figs. 3 and 5 lack necessary statistical analysis.
  3. Section 3.3. Tests of the advance of infection on pomegranate fruits. It is suggested that the authors add the pictures of gray mold disease on pomegranate fruits.
  4. Section 3.8. Figure 10. The author only compared the quantity of plant cell wall degrading enzymes in strains BcPgIs-1, BcPgIs-3 and MIC, and did not analyze the unique enzymes in each of strains. The relationship between enzyme composition, distribution and enzyme activity should be supplemented.
  5. Table 3. Orthologous genes associated with pathogenicity identified in the genomes of three B. cinerea strains. Are the pathogenicity-related genes of these strains the same? Although all these strains contain these pathogenicity-related genes, their virulence is different. The author should supplement the relationship between disease-related genes and the virulence of the strains.

Author Response

In response to the comments and suggestions regarding the manuscript entitled “Comparison and analysis of the genomes of three strains of Botrytis cinerea isolated from pomegranate” submitted to Microorganisms, in the Special Issue

Feature Papers in Plant–Microbe Interactions in North America, the following aspects are discussed:

Reviewer 1

Initial comment:

Botrytis cinerea is an important necrotrophic fungal pathogen that infects a wide variety of plants and causes gray mold disease. The current manuscript reported a comparison and analysis of the genomes of three strains of B. cinerea isolated from pomegranate. The results obtained could be useful information for researchers who worked in the similar field. In general, this is an interesting study.

Response:

Thank you very much for your comment

Comment 1:

Section 2.3. Generally, when measuring enzyme activity, it is necessary to define enzyme units. For example, one enzyme unit may be defined as the amount of enzyme that produces a certain quantity of product per minute under specific conditions, and so on. The “activity halo” can provide a qualitative assessment of enzyme activity, but it is not accurate for quantitative analysis. The authors should recalculate the enzyme activity. 

Response:

The tests to measure enzymatic activity were carried out on a plate, allowing the determination of the power index, which relates the diameter of the activity halo to the diameter of the colony growth (Line 136). In this case, it is not possible to determine units of enzymatic activity because the release of product or the consumption of substrate is not measured. At first glance, the test seems qualitative. However, by calculating the potency index, a comparison can be made to identify which strains produce more of the enzyme, evident through the halo of substrate degradation in the case of cellulase and pectinase, or the oxidation of ABTS by laccase activity.

Comment 2:

Figs. 3 and 5 lack necessary statistical analysis.

Response:

The statistical analysis for Figs. 3 and 5 was conducted by comparing the data collected each day (lines 150-152, 289-291, 325-327).

Comment 3:

Section 3.3. Tests of the advance of infection on pomegranate fruits. It is suggested that the authors add the pictures of gray mold disease on pomegranate fruits.

Response:

There is an error. The abstract and section 3.3 mention that the BcPgIs-3 and MIC strains had completely invaded the pomegranate fruits. This is incorrect; the fungus does not invade the fruit. Instead, what is being quantified is the progression of the disease according to the diagrammatic scale reported by Patricio-Hernández et al. in 2023 (https://doi.org/10.18781/r.mex.fit.2302-9). This was corrected in the document (lines 26-27 and 277-278), and section 2.4 (lines 138-147). We cannot add photographs because only data on disease progression were collected using the diagrammatic scale. We only have a photo of a fruit with 100% damage caused by the MIC strain, and we do not think it’s appropriate to include just one image without controls and other strains. We believe the results in Figure 5 are sufficient to illustrate the effect of the strains on disease progression in pomegranate fruits.

Comment 4:

Section 3.8. Figure 10. The author only compared the quantity of plant cell wall degrading enzymes in strains BcPgIs-1, BcPgIs-3 and MIC, and did not analyze the unique enzymes in each of strains. The relationship between enzyme composition, distribution and enzyme activity should be supplemented.

Response:

An analysis was conducted on each gene involved in the degradation of the polymers that make up the plant cell wall, and we found that some were unrelated to the desired activity, so they were removed. As a result, Figure 10 was updated. Four tables were added as supplementary material, displaying the enzymes encoded by the identified genes. The results describe the differences found in the composition and distribution of the enzymes (Line 379-393). In the discussion, it is mentioned that the high pectinolytic activity of the MIC strain could be because it has a greater number of genes related to pectin degradation (line 530-532).

Comment 5:

Table 3. Orthologous genes associated with pathogenicity identified in the genomes of three B. cinerea strains. Are the pathogenicity-related genes of these strains the same? Although all these strains contain these pathogenicity-related genes, their virulence is different. The author should supplement the relationship between disease-related genes and the virulence of the strains.

Response:

The genes listed in Table 3 are orthologs to genes previously identified in other B. cinerea strains and have also been linked to virulence or fungal development. Furthermore, following the suggestion of the second reviewer, these sequences were compared with the PHI-base database, showing that all of these genes play a role in virulence and/or pathogenicity. Although the same genes were found across all three strains, we discuss in the paper the possible processes that might contribute to the observed differences in virulence (line 584-587).

Corrections are highlighted in yellow in the manuscript.

Your comments and suggestions will contribute significantly to improving this manuscript.

Thank you for your observations,

Dr. Yuridia Mercado-Flores

Reviewer 2 Report

Comments and Suggestions for Authors

The comparative analysis of three strains of Botrytis cinerea presents a well-documented study, highlighting their morpho-physiological characteristics alongside phylogenomic analysis. Overall, the experiments are clearly described, with the exception of a few areas where further discussion is warranted. Additionally, I have suggested some further analyses that could enhance the comprehensiveness of the study.

It would be beneficial to perform PHI-base categorization of the identified genes to compare virulence-related genes. Similarly, analyzing genes encoding secondary metabolites using the antiSMASH database would add further depth to the study.

The abstract should include more specific information related to the genomic data obtained from the strains. Generic statements such as “genomes demonstrated robustness and consistency” can be omitted and replaced with details on the variation in effector repertoires and secondary metabolite clusters.

In the introduction, consider including a brief explanation of how conidia and sclerotia infect healthy plants.

The sentence “current studies and genomic analysis…” is vague in its reference to previous comparative genomics studies. It would be more informative to elaborate on the genomic structure and functional diversity observed among B. cinerea isolates.

It is also important to deposit all sequence data in a public sequence repository to ensure accessibility.

As the growth rate of BcPgls-3 appears to be highest due to rapid conidia production, it would be helpful to assess conidia counts at later time points, when all strains reach their growth saturation.

How was the percentage of disease progression calculated? In its current form, a 100% disease progression implies that the entire fruit is infected by 9 days post-inoculation (dpi). Clarification of this metric is needed.

For the GO analysis, it would be more informative to include the number of genes from each strain associated with each GO category. This would provide deeper insight into strain-specific gene functions.

The discussion section requires more detailed interpretation of the findings. Specific elements of the comparative genomics, such as the Circos and Venn diagrams, need further elaboration. There is limited discussion on the observed variation across strains and its potential implications, which should be addressed more thoroughly.

Author Response

In response to the comments and suggestions regarding the manuscript entitled “Comparison and analysis of the genomes of three strains of Botrytis cinerea isolated from pomegranate” submitted to Microorganisms, in the Special Issue

Feature Papers in Plant–Microbe Interactions in North America, the following aspects are discussed:

Reviewer 2

Initial comment:

The comparative analysis of three strains of Botrytis cinerea presents a well-documented study, highlighting their morpho-physiological characteristics alongside phylogenomic analysis. Overall, the experiments are clearly described, except for a few areas where further discussion

Response:

Thank you very much for your comments, which are addressed below.

Comment 1:

It would be beneficial to perform PHI-base categorization of the identified genes to compare virulence-related genes. Similarly, analyzing genes encoding secondary metabolites using the antiSMASH database would add further depth to the study.

Response:

The analysis was performed using antiSMASH (lines 244-245). Although clusters for synthesizing pharmacologically important secondary metabolites were identified, no genes associated with the production of phytotoxic compounds that could act as pathogenicity and virulence factors were found (lines 398-399).

Furthermore, the genes listed and described in Table 3 were analyzed using the PHI-base database (lines 242-243), confirming that each of them may be related to pathogenicity and virulence processes. Table S5, which includes the results of this analysis, is provided as supplementary material (lines 396-397).

Comment 2:

The abstract should include more specific information related to the genomic data obtained from the strains. Generic statements such as “genomes demonstrated robustness and consistency” can be omitted and replaced with details on the variation in effector repertoires and secondary metabolite clusters.

Response:

The phrase "genomes demonstrated robustness and consistency" was replaced with details about the sequencing results (lines 27-29). Regarding variation in effector repertoires, we believe the abstract already mentions it. We do not include results for secondary metabolites because the analysis with the current antiSMASH tool did not identify clusters for the synthesis of phytotoxic compounds.

Comment 3:

In the introduction, consider including a brief explanation of how conidia and sclerotia infect healthy plants.

Response:

The introduction was enhanced with a brief explanation of how conidia and sclerotia infect healthy plants (lines 63-69).

Comment 4:

The sentence “current studies and genomic analysis…” is vague in its reference to previous comparative genomics studies. It would be more informative to elaborate on the genomic structure and functional diversity observed among B. cinerea isolates.

Response:

The sentence “current studies and genomic analyses…” was removed to prevent confusion, and the paragraph was adjusted (line 70-74).

Comment 5:

It is also important to deposit all sequence data in a public sequence repository to ensure accessibility.

Response:

The genome sequences were deposited in the GenBank database, and the access codes are provided in the manuscript (lines 301-303).

Comment 6:

As the growth rate of BcPgls-3 appears to be highest due to rapid conidia production, it would be helpful to assess conidia counts at later time points, when all strains reach their growth saturation.

Response:

In this case, we identified an error. For all media, the plates were incubated for 10 days, during which all strains invaded the Petri dish. This was the period when the conidia count was performed. The manuscript was revised to include this change (lines 107, 110, 262).

Comment 7:

How was the percentage of disease progression calculated? In its current form, a 100% disease progression implies that the entire fruit is infected by 9 days post-inoculation (dpi). Clarification of this metric is needed.

Response:

The percentage of disease progression was determined using a diagrammatic scale. This information was included in the document (lines 138-147).

Comment 8:

For the GO analysis, it would be more informative to include the number of genes from each strain associated with each GO category. This would provide deeper insight into strain-specific gene functions.

Response:

Figure 7 shows the number of genes linked to each GO category, which are listed above each bar. For clarity, this information is also included in the figure caption (line 472).

Comment 9:

The discussion section requires more detailed interpretation of the findings. Specific elements of the comparative genomics, such as the Circos and Venn diagrams, need further elaboration. There is limited discussion on the observed variation across strains and its potential implications, which should be addressed more thoroughly.

Response:

The discussion section (lines 566-674) explains the differences observed regarding what was requested.

Corrections are highlighted in green in the manuscript.

Your comments and suggestions will contribute significantly to improving this manuscript.

Thank you for your observations,

Dr. Yuridia Mercado-Flores

Round 2

Reviewer 1 Report

Comments and Suggestions for Authors

The authors have revised the manuscript according to the reviewers' suggestions. Overall, this is an interesting paper that serves as a valuable reference for researchers in this field.

Author Response

Thank you for your time and effort.

Reviewer 2 Report

Comments and Suggestions for Authors

I am happy with the revised version.

Author Response

Thank you for your time and effort.